# CLONE DETERMINISTIC 3D WORLDS WITH GEOMETRICALLY-REGULARIZED WORLD MODELS

## ABSTRACT

A world model is an internal model that simulates how the world evolves. Given past observations and actions, it predicts the future physical state of both the embodied agent and its environment. Accurate world models are essential for enabling agents to think, plan, and reason effectively in complex, dynamic settings. However, existing models remain fragile, lack robustness, and struggle with reliable long-horizon predictions. In this work, we take a step toward building a truly accurate world model by addressing a fundamental yet open problem: constructing a model that can fully clone and overfit to a deterministic 3D world. Exteroceptive sensory inputs, such as images, are high-dimensional and encode complex, nonlinear mappings from the underlying physical processes, making precise latent-state prediction challenging. Overcoming this requires a representation space that faithfully captures the underlying physical states while minimizing information loss and noise. Such a representation simplifies the subsequent dynamics modeling task, making representation quality critical to overall world model accuracy. We propose Geometrically-Regularized World Models (GRWM), which enforces that consecutive points along a natural sensory trajectory remain close in latent space. This approach yields significantly improved latent representations that align closely with the true topology of the environment. Our method is plug-and-play, requires only minimal architectural modification, and scales naturally with trajectory length. It applies broadly across latent generative backbones and achieves state-of-the-art fidelity on long-horizon prediction benchmarks. Both qualitative and quantitative analyses show that its success comes from learning a latent space with superior geometric structure.

## 1 INTRODUCTION

Creating a high-fidelity, interactive clone of an environment purely from observational data has long been a central ambition in artificial intelligence. Such a clone can serve as a simulator for reinforcement learning agents (Hafner et al., 2020; Hao et al., 2025), enable task planning in robotics (Mendonca et al., 2023), and facilitate controllable content generation in games (Alonso et al., 2024; Quevedo et al., 2024). World models are the primary tools for achieving this goal: they aim to capture an environment's dynamics, predict its future states, and simulate its evolution (Sutton, 1991; Ha & Schmidhuber, 2018; Schrittwieser et al., 2020; LeCun, 2022).

Most current world models focus on open-world settings (Quevedo et al., 2024; He et al., 2025; Ball et al., 2025), generating unconstrained and often random environments in which each simulation produces a different world. This approach is intended to enhance generalization by exposing agents to diverse training scenarios. However, the emphasis on randomness can lead to unstable dynamics, making such models ill-suited for applications that demand reliable prediction and precise planning in fixed tasks. In these cases, they often produce futures that are merely plausible, not faithful.

Achieving precise world simulation with world models requires overcoming two tightly intertwined challenges:

- Representation learning. Exteroceptive sensory data, such as images, are high-dimensional and encode complex, nonlinear mappings from underlying physical processes. This makes accurate future-state prediction difficult even under full observability. For example, a simple spatial translation in the physical world can correspond to a highly nonlinear trajectory in pixel space. Addressing this requires a representation space that faithfully encodes the underlying physical states while minimizing information loss and noise, thereby simplifying the subsequent dynamics modeling task.
- Dynamics modeling. Even with an optimal representation, the dynamics model must capture a broad range of transition patterns — including 3D transformations, logical rules,

causal dependencies, and temporal memory. The challenge lies in building a unified mechanism that can accurately model all these regularities.

These challenges are inherently coupled: a poor representation forces the dynamics model to operate in a noisy, entangled latent space, increasing prediction complexity and reducing generalization; conversely, a well-structured representation is of limited utility if the dynamics model cannot capture the full spectrum of transition patterns. Progress toward generic neural world simulation therefore demands a co-design of representation learning and dynamics modeling, ensuring that the latent space is both physically meaningful and optimally aligned with the predictive capabilities of the dynamics model.

In this work, we focus on the faithful cloning of deterministic 3D environments — settings governed by fixed rules, such as a 3D maze with a static map. We target these environments for three reasons: (1) Their consistent dynamics make them amenable to precise predictive modeling. (2) The absence of stochasticity eliminates uncertainty, enabling rigorous evaluation of a world model's fidelity. (3) Many important real-world applications — including robot navigation in fixed spaces and game AI operating on static maps — inherently involve deterministic environments. In such cases, the objective is not to produce a plausible world, but to reproduce the unique, true trajectory of the environment. Ultimately, our goal is to construct a digital twin indistinguishable from the original in both rules and behavior.

Yet, we find that achieving accurate long-horizon cloning of even simple deterministic 3D environments remains an open challenge. Across all state-of-the-art baselines we evaluated, none were able to maintain fidelity over extended horizons: small prediction errors accumulate rapidly, causing trajectories to diverge from reality after only a few steps. In contrast, when the dynamics model is provided with the environment's underlying physical states — rather than high-dimensional exteroceptive signals such as images — it can produce remarkably accurate long-horizon predictions. This observation suggests that the effectiveness of a world model is fundamentally constrained by the structure of its latent representation space. This raises the central question: How can we build a self-supervised representation that aligns with the underlying physical states, enabling stable and accurate long-horizon predictions?

While most representation learning methods are trained on IID datasets, real-world robotic systems naturally acquire continuous sensory trajectories through interaction. These trajectories inherently encode geometric and temporal regularities that can be leveraged to improve representation quality(Földiák, 1991; Wiskott & Sejnowski, 2002; Goroshin et al., 2015; Chen et al., 2018). Recent work(Wang et al., 2024) has shown that applying geometric regularization to temporal trajectories can substantially improve the topological structure of a latent space for 3D objects, benefiting tasks such as semantic classification and pose estimation. We extend this idea to the domain of 3D environments, demonstrating that trajectory-based geometric regularization can significantly enhance latent representations and, in turn, markedly improve the effectiveness of world models.

This work aims to bridge the gap to oracle-level performance through unsupervised representation learning. We introduce Geometrically-Regularized World Models (GRWM) — a framework that learns a latent space mirroring the geometry of the true state manifold, without requiring access to the ground-truth states. At its core is a lightweight geometric regularization module that can be seamlessly integrated into standard autoencoders, reshaping their latent space to provide a stable foundation for effective dynamics modeling. By focusing on representation quality, GRWM offers a simple yet powerful pipeline for systematically improving world model fidelity.

In summary, our main contributions are as follows:

1. We formalize the problem of high-fidelity cloning of deterministic environments, shifting the focus from open-world generation to reproducible fidelity, and curate dedicated datasets for this task.

2. We introduce Geometrically-Regularized World Models (GRWM), a general, unsupervised geometric regularization method that enhances representation quality. This approach validates the "representation matters" hypothesis by demonstrating that a well-structured latent space systematically improves the performance of various dynamics models without altering their architecture.

3. We demonstrate that by focusing on representation, GRWM achieves state-of-the-art performance in long-horizon trajectory prediction. Extensive qualitative and quantitative analyses validate its success stems from learning a superior geometric structure in the latent space.

## 2    PRELIMINARY

**The Next-State Prediction Problem.** World models aim to perform *next-state prediction*: learning a function that predicts future observations given the current state and an action. An environment evolves through latent states $s_t$, which yield observations $o_t$ after actions $a_t$. In the partially observable case, $o_t$ gives only partial information about $s_t$, requiring integration of past history to form a belief state. In the fully observable case, $s_t \approx o_t$, yet predicting in observation space remains difficult because simple physical transitions (e.g., translation) correspond to complex, non-linear changes in high-dimensional pixel space. This necessitates a learned representation space that can linearize these dynamics.

Our objective is to learn a world model $M$ that can faithfully clone a deterministic environment from purely observational data. The model is trained on a dataset $\mathcal{D}$ consisting of trajectories $\tau = \{(o_1, a_1), \ldots, (o_T, a_T)\}$, where $o_t$ is an observation (image) and $a_t$ is an action at timestep $t$. The deterministic environment assumption states that, for any initial state and sequence of actions, there is exactly one resulting sequence of observations. The fidelity of the learned model is assessed through its ability to generate long-horizon rollouts. Specifically, given a starting observation $o_1$ and an action sequence $\{a_t\}_{t=1}^{T}$, the model produces a rollout $\{\hat{o}_t\}_{t=1}^{T}$. At each timestep $t$, we compute the frame-wise Mean Squared Error (MSE) between the generated and ground-truth observations: $\text{MSE}(t) = \|o_t - \hat{o}_t\|_2^2$. This produces an error curve $\{\text{MSE}(t)\}_{t=1}^{T}$ that reveals how prediction error accumulates over time.

**Latent Generative Models.** We adopt a latent generative framework consisting of two stages: (1) an autoencoder that learns a compressed latent representation (Kingma & Welling, 2013; Higgins et al., 2017), and (2) a generative model that captures the dynamics in that latent space (Ha & Schmidhuber, 2018; Hafner et al., 2019; Bruce et al., 2024). Our work focuses on the representation stage, as its quality critically determines the downstream rollout performance.

**Contrastive Learning.** Pure reconstruction objectives often yield degenerate representations that ignore temporal cues. Contrastive learning principles (Hadsell et al., 2006; van den Oord et al., 2018; Chen et al., 2018; 2020; He et al., 2020; Wang & Isola, 2020; Yeh et al., 2022; Garrido et al., 2022; Wang et al., 2024) suggest a remedy: enforce similarity for related samples and repulsion for unrelated ones. We extend this idea temporally, targeting the specific aliasing and stability challenges of world modeling.

## 3    GEOMETRICALLY-REGULARIZED WORLD MODELS

### 3.1    MOTIVATION: REPRESENTATION MATTERS

To motivate our focus on representation learning, we first conduct a revealing experiment on the Maze 3x3 environment. We establish an "oracle" world model by training a world model directly on the ground-truth underlying states (i.e., the agent's coordinates). As shown in Figure 1, this oracle, which bypasses the challenge of perception, predicts future states with near-perfect accuracy. This demonstrates that high-fidelity cloning is feasible if a perfect representation is available.

However, when the same dynamics model is trained on the latent space of a standard VAE, its performance collapses. This gap highlights that the primary bottleneck is not the dynamics model, but the representation itself. The VAE's latent space, optimized solely for reconstruction, is not structurally aligned with the environment's state manifold, leading to catastrophic errors. As a preview, Figure 1 also shows that a regularized VAE, trained with the geometric priors we introduce next, significantly closes this performance gap. This experiment solid-

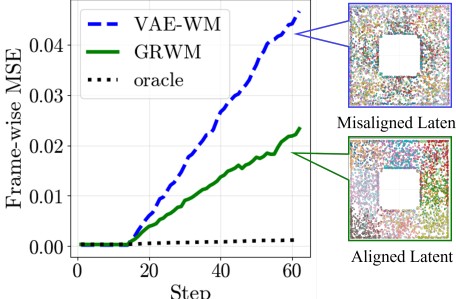

Figure 1: **Representation quality is the primary bottleneck for long-horizon prediction.** Frame-wise MSE on the Maze 3x3 dataset. **(Left)** An oracle model using ground-truth states (black dotted) achieves near-zero error, establishing a performance upper bound. In contrast, a standard VAE-based world model (blue dashed) accumulates error rapidly. Our GRWM (green solid) significantly closes this gap by learning a more structurally aligned latent space **(Right Bottom)**, while the VAE's representation remains disorganized **(Right Top)**. For further details, see Section 4.

ifies our central thesis: high-fidelity cloning requires a latent space that is structurally aligned with the environment's true underlying state manifold.

Building on the insight, we propose Geometrically-Regularized World Models, which consists of two key components: (1) an architecture that incorporates temporal context, and (2) a regularization loss that explicitly enforces structure in the latent space. We describe each component in the following subsections.

## 3.2 Temporal Contextualize Architecture

Learning underlying states from a single frame is difficult due to *perceptual aliasing*, where distinct states can yield nearly identical observations. For example, two different positions in a maze may appear visually indistinguishable. To resolve this ambiguity and capture the true dynamics of the environment, the model requires temporal context.

We design the representation model with a causal encoder $E$ and an instantaneous decoder $D$. The encoder maps a sequence of recent observations $(o_{t-k}, \ldots, o_t)$ to a latent representation $z_t$, which summarizes the information necessary to infer the current state. The decoder $D$ then reconstructs only the current observation $\hat{o}_t$ from $z_t$:

$$z_t = E(o_{t-k}, \ldots, o_t), \quad \hat{o}_t = D(z_t).$$

This design ensures that $z_t$ is a compact representation of the present state, enriched by past context. To instantiate this design, we build upon variational autoencoder framework augmented with temporal aggregation.

**VAE with Temporal Aggregation.** Each frame is first encoded independently with a 2D CNN. The resulting frame-level features are then aggregated by a causal Transformer with a sliding temporal window, ensuring that $z_t$ only attends to a limited range of past frames up to and including time $t$. This windowed design captures short-term temporal context while maintaining causality.

## 3.3 Temporal Contrastive Regularization

While the causal encoder introduces temporal context, the standard reconstruction objective on the final frame is insufficient to guarantee a well-structured latent space. The model might learn a "lazy" solution, ignoring the context and relying solely on the last frame. Inspired by principles from contrastive representation learning, we introduce a temporal contrastive regularization loss to explicitly regularize the latent space.

The output of the encoder, a sequence of latent vectors $\mathbf{z} \in \mathbb{R}^{B \times L \times \cdots}$, is first passed through a projection layer (a linear layer) to produce embeddings $\mathbf{p} \in \mathbb{R}^{B \times L \times D}$ (Chen et al., 2020). These embeddings are then $L_2$-normalized to lie on the unit hypersphere: $\mathbf{p}' = \frac{\mathbf{p}}{\|\mathbf{p}\|_2}$ (Wang & Isola, 2020). Our regularization objectives are applied to these normalized embeddings, $\mathbf{p}'$.

**Temporal Slowness Loss ($\mathcal{L}_{\text{slow}}$).** The idea of temporal slowness is that consecutive or nearby states in a trajectory should have similar latent representations, reflecting the intuition that the underlying state of the environment evolves gradually over time (Wiskott & Sejnowski, 2002). Our loss encourages all pairs of frames within the same trajectory's context window to be close to one another on the hypersphere. This enforces that the entire trajectory segment is mapped to a compact and continuous path in the representation space, ensuring that the latent representation evolves slowly and smoothly over time. We formalize this by minimizing the average L2 distance between all pairs of embeddings within a trajectory:

$$\mathcal{L}_{\text{slow}} = \mathbb{E}_{b \sim \mathcal{D}} \left[ \mathbb{E}_{(\mathbf{p}'_i, \mathbf{p}'_j) \sim \mathcal{P}'_b \times \mathcal{P}'_b} \left[ \left\| \mathbf{p}'_i - \mathbf{p}'_j \right\|_2 \right] \right],$$

where $P'_b = \{\mathbf{p}'_{b,t}\}_{t=0}^{L-1}$ is the set of normalized embeddings for a trajectory $b$.

**Latent Uniformity Loss ($\mathcal{L}_{\text{uniform}}$).** Slowness alone can lead to feature collapse (i.e., the model maps many inputs to a small region of the latent space). The uniformity loss mitigates this issue by encouraging embeddings to distribute evenly on the hypersphere. It is formally expressed as:

$$\mathcal{L}_{\text{uniform}} = \log \mathbb{E}_{(\mathbf{p}'_i, \mathbf{p}'_j) \sim \mathcal{P}_{\text{neg}}} \left[ e^{-2\left\| \mathbf{p}'_i - \mathbf{p}'_j \right\|_2^2} \right],$$

where $\mathcal{P}_{neg}$ is the distribution of all pairs of embeddings from different trajectories in the batch.

**Overall Training Objective.** The complete autoencoder is trained end-to-end by minimizing a total objective function that combines the reconstruction loss, a KL-divergence term from the VAE framework, and our two proposed regularization terms. The final loss is:

$$\mathcal{L}_{total} = \mathcal{L}_{recon} + \beta\mathcal{L}_{KL} + \lambda_{slow}\mathcal{L}_{slow} + \lambda_{uniform}\mathcal{L}_{uniform}$$

where $\beta$, $\lambda_{slow}$, and $\lambda_{uniform}$ are hyperparameters that balance the contribution of each term.

### 3.4 PRACTICAL CONSIDERATIONS

GRWM is designed to close the performance gap to an oracle model in a simple and general way, without requiring supervised data. Our geometric regularization module embodies this principle. Its implementation is minimalist, requiring only two additions to a standard VAE framework: a lightweight projection head and our two supplementary loss terms. This design makes GRWM a plug-and-play component that can be applied to enhance any latent generative model without needing to alter its core architecture. The computational overhead is minimal and scales naturally with batch size and trajectory length, without complex sampling or augmentation strategies. This simplicity and generality make it a practical tool for improving representation quality, turning any standard autoencoder into a powerful foundation for high-fidelity world modeling.

## 4 EXPERIMENTS

### 4.1 SETUP

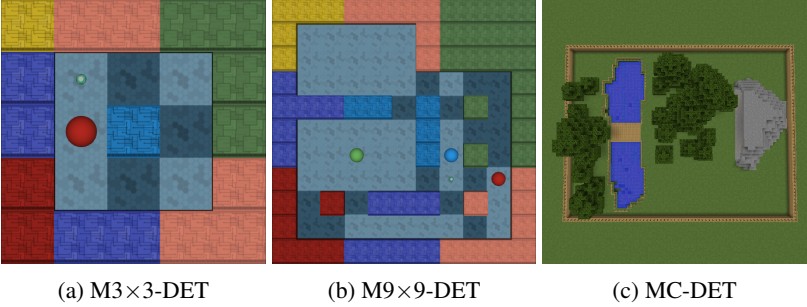

(a) M3×3-DET          (b) M9×9-DET          (c) MC-DET

Figure 2: Top-down visualizations of our three closed environments: M3×3-DET, M9×9-DET, and MC-DET. These maps illustrate the overall layout and are for visualization purposes only; they are not provided as input to the agent. The agent's input is restricted to first-person observations. For a more representative depiction of the agent's surroundings, high-angle perspective views are also included in Appendix Figure 11, offering a better sense of the environments' three-dimensional structure and scale.

**Datasets.** We introduce three datasets collected in deterministic environments: Maze 3×3-DETERMINISTIC (M3×3-DET), Maze 9×9-DETERMINISTIC (M9×9-DET), and Minecraft-DETERMINISTIC (MC-DET). Trajectories in these datasets are sequences of (action, observation) pairs from a first-person perspective. The map layout for each environment is fixed, rendering the trajectories fully deterministic. Figure 2 shows top-down views of these environments for visualization; these maps are not available to the agent. The Maze datasets differ in size and complexity, while the Minecraft dataset provides richer visual observations. Further details on data collection are in Appendix E.

**Baselines.** Our world model framework is flexible regarding the choice of dynamics model. For a fair comparison, we select three popular dynamics models: Standard Diffusion (SD) (Alonso et al., 2024), Video Diffusion Models (VD) (Ho et al., 2022), and Diffusion Forcing (DF) (Chen et al., 2024)[1]. For each of these dynamics models, we train and compare two versions: a vanilla version that uses a standard VAE, and GRWM version. This direct comparison allows us to isolate the

---

[1]If the original paper does not support actions explicitly, we treat the action as an additional condition and feed it into the network.

contribution of representation learning and assess its impact across different dynamics architectures. A detailed description of training configurations is provided in Appendix D.

**Metrics.** We evaluate prediction fidelity using *frame-wise MSE*. Since the environments are deterministic, a unique ground-truth future exists for any given initial state and action sequence. We can therefore compute the mean squared error between the predicted and ground-truth observations at each timestep in pixel space. This metric reflects the step-by-step discrepancy between predicted and actual observations across the entire trajectory. A lower frame-wise MSE indicates higher fidelity in cloning the environment's dynamics over time.

## 4.2 ROLLOUT FIDELITY

To quantitatively evaluate rollout prediction fidelity, we present our main results in Figure 3. The results demonstrate a consistent advantage for our method, and the performance gap widens with rollout length. Our method (solid lines) maintains a significantly lower prediction error compared to the baseline using a vanilla VAE (dashed lines). Baseline models accumulate error quickly, causing trajectories to diverge, whereas our method maintains a much flatter error curve. This shows stronger long-term temporal consistency.

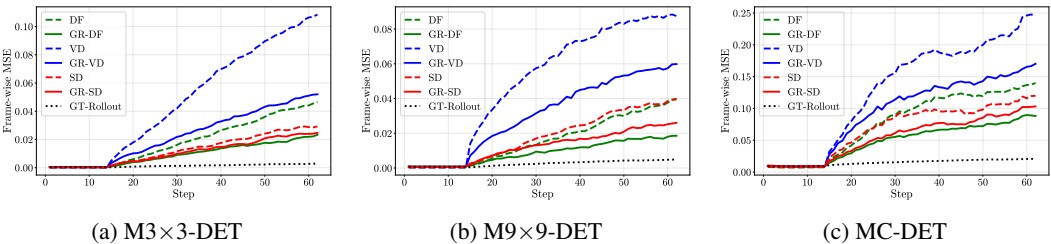

(a) M3×3-DET  (b) M9×9-DET  (c) MC-DET

Figure 3: Rollout Performance. Frame-wise MSE between predicted and ground-truth trajectories on (a) M3x3-DET, (b) M9x9-DET, and (c) MC-DET datasets. The oracle model (black dotted line), which operates on the true underlying states, establishes a lower bound on error. For all three dynamics models—Diffusion Forcing (DF), Video Diffusion (VD), and Standard Diffusion (SD)—our GRWM (solid lines) consistently outperforms baselines (dashed lines), demonstrating significantly lower error accumulation over 63 steps and substantially closing the performance gap to the oracle.

These findings confirm our central hypothesis: by learning a more structured and aligned latent space, our method provides a stronger foundation for the dynamics model, leading to substantial improvements in high-fidelity prediction.

## 4.3 QUALITATIVE ANALYSIS OF LONG-HORIZON GENERATION

To further probe the long-term stability, we conduct an extreme long-horizon generation task. For this experiment, we specifically select our best-performing dynamics model, Diffusion Forcing[2], and the Maze 9x9-CE dataset. This environment is particularly well-suited for this analysis due to its high degree of perceptual aliasing—it contains many corridors and rooms that are visually similar, making it difficult for a model to distinguish between different states based on a single frame. We use a sequence of randomly sampled actions to simulate an exploratory trajectory.

We tasked the model with producing 10,000 consecutive frames from a single starting point. We then visualize these long trajectories by sampling and displaying frames at a fixed interval of every 1000 steps, as shown in Figure 4 and Figure 5. Additional results under different starting conditions are provided in the Appendix C.

The results reveal a critical failure mode in the baseline model. As seen in the figures, the VAE-WM quickly succumbs to mode collapse, getting trapped in repetitive loops that render nearly identical, low-complexity frames. Our interpretation is that the model learns to "teleport" between visually similar but causally disconnected regions of the environment. For example, it can spend thousands of consecutive frames generating views of a single-colored wall (e.g., the pink wall in Figure 4; the green and blue walls in Figure 5). The pixel-based reconstruction loss forces the model to map

---

[2]In the following discussion, we refer to the VAE combined with diffusion as **VAE-WM**, and our method with Diffusion Forcing as **GRWM**.

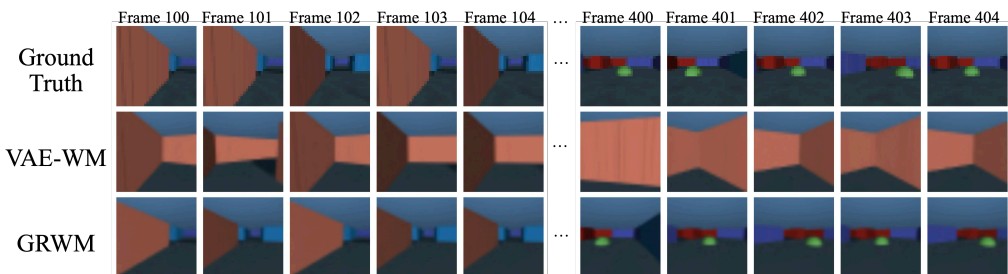

Figure 4: Qualitative comparison of medium-horizon rollouts. We visualize consecutive frames around frame 100 and frame 400. Our method (GRWM) maintains high similarity to the ground truth throughout, while the baseline VAE-WM gets trapped near the pink wall, indicating that VAE-WM tends to "teleport" between visually similar but distinct locations.

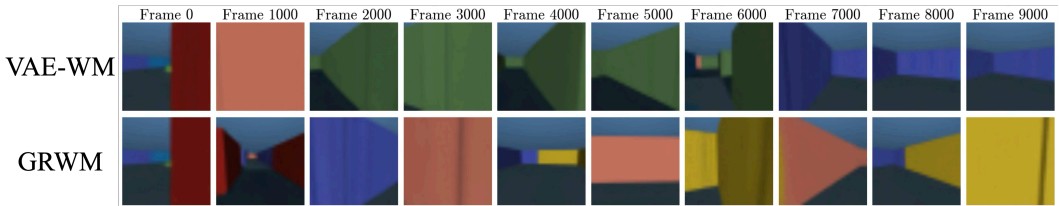

Figure 5: Qualitative comparison of ultra long-horizon rollouts on the Maze 9x9-CE dataset. Frames are sampled every 1000 steps from a 10,000-step rollout. The baseline VAE-WM frequently gets stuck generating the same color states, failing to explore the environment effectively. In contrast, GRWM produces a coherent and diverse trajectory, successfully exploring different regions while preserving long-term temporal and structural consistency.

visually similar observations—such as different walls of the same color—to nearby points in the latent space, irrespective of their true distance or causal connection within the environment. This creates "attractor states" and an entangled manifold. The dynamics model, operating in this flawed space, learns that jumping between these close latent points is a low-cost action, resulting in the observed "teleportation" between safe havens of low reconstruction error instead of navigating the true, complex topology of the world.

In contrast, our GRWM generates a far more coherent and diverse trajectory because its representation is regularized to be consistent with the true underlying state manifold. As shown in Figure 4, GRWM maintains high fidelity at both 100 and 400 frames, while the VAE-WM fails. In the longer rollout (Figure 5), the sequence of frames from GRWM clearly shows movement and exploration. The appearance of the yellow wall in later frames is not a random plausible image; it is evidence of a continuous traversal through a latent space that mirrors a physically possible path in the environment. Our geometric regularization forces the model to respect the world's structure, preventing the state-skipping and teleportation artifacts that dominate the VAE rollouts.

### 4.4 LATENT REPRESENTATION ANALYSIS

**Latent Probing.** We perform a latent probing analysis to quantitatively assess how well the learned representations capture the true underlying states of the environment (i.e., agent position $(x, y)$ and orientation $\theta$). We freeze the trained autoencoder and use its encoder to obtain latent vectors for all observations. A small MLP probe is then trained to predict the ground-truth states from these latent vectors. We report regression MSE on a held-out validation set, where a lower value indicates that the latent space is more informative and better aligned with the environment's true state manifold.

GRWM consistently learns latent representations that are more linearly predictive of the ground-truth agent state. The results, summarized in Table 1, clearly support our hypothesis. Across all three datasets, our method leads to a significant reduction in regression MSE. Notably, the improvement

Table 1: Latent probing analysis. GRWM consistently learns representations that are more predictive of the true underlying states. We report regression MSE of an MLP probe on a held-out set (lower is better).

| Model | M3×3-DET | M9×9-DET | MC-DET |
|---|---|---|---|
| VAE-WM | 0.082 | 0.106 | 0.137 |
| GRWM | 0.031 | 0.058 | 0.081 |

is consistent regardless of the environment's complexity, highlighting the general applicability and effectiveness of our approach in structuring the latent space.

**Latent Clustering.**   To further investigate the structure of the learned representations, we conduct a clustering analysis. We first obtain latent vectors for a set of frames using the trained encoder and then apply a k-means algorithm (with $k = 20$ clusters) to group these vectors in the latent space. To visualize the result, we plot each frame as a point at its ground-truth $(x, y)$ position within the environment and assign it a color based on its latent cluster ID. The results are shown in Figure 6.

GRWM successfully forces the model to learn a latent manifold that is structurally aligned with the environment's true topology. The baseline VAE (top row) produces noisy and fragmented clusters. A single color (representing a single latent cluster) is scattered across disparate and often distant regions of the map. This indicates a highly entangled representation, where frames from fundamentally different underlying states are incorrectly mapped to the same region of the latent space. Such a representation provides a poor foundation for a dynamics model, as it fails to distinguish between causally distinct states. In contrast, our method (bottom row) produces remarkably coherent and spatially contiguous clusters. Each color largely corresponds to a distinct, localized region of the environment, such as a specific corridor or room. By correctly grouping states that are close in the physical world, our method provides a smooth and well-structured landscape upon which a dynamics model can learn accurate and generalizable transitions.

## 5 ABLATION STUDIES

We conduct ablation studies to validate the contribution of our core components and design choices. Specifically, we examine four aspects: (1) the necessity of the two core loss terms, (2) the role of the projection head, (3) the impact of latent dimension, and (4) the effect of critical design choices on model performance. The detailed results are provided in Appendix B.

## 6 DISCUSSION AND CONCLUSION

In this work, we addressed the challenge of high-fidelity cloning for deterministic, closed environments. We began from a simple yet powerful premise: representation matters. While much of the field has focused on developing more powerful dynamics models, our work makes a definitive case that the primary bottleneck to long-horizon prediction is the quality of the latent space in which those dynamics operate. We introduced Geometrically-Regularized World Models (GRWM), a framework that learns a latent space structurally aligned with the environment's true state manifold through a combination of a temporal-contextual architecture and a novel geometric regularization loss. Our experiments consistently demonstrated that by improving the representation, we could systematically enhance the performance of various dynamics models, significantly reducing error accumulation and preventing the catastrophic trajectory divergence that plagues standard methods.

**Limitations and Future Vision.**   While our geometric regularization substantially improves latent space topology, it alone does not guarantee that the learned representation is equivalent to the true, irreducible physical state. Any residual misalignment, however small, could eventually compound over extremely long horizons. This limitation points toward the ultimate goal and grand challenge for the field: the unsupervised discovery of the true underlying generative factors of an environment.

An optimal representation should capture these physical states, reducing high-dimensional sensory signals to their simplest, most informative form without losing information critical for prediction. Recent advances in self-supervised learning, vector quantization, and learned compression have

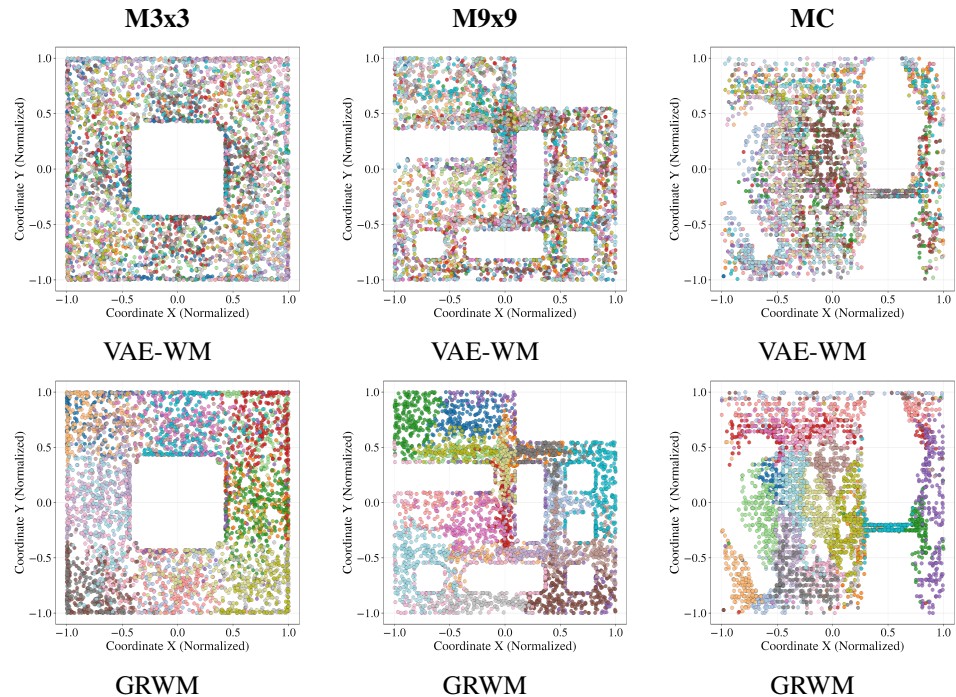

Figure 6: Visualization of latent space structure through clustering analysis. We perform k-means clustering ($k = 20$) on the latent representations of frames. Each point in the plots corresponds to a frame, positioned according to its true $(x, y)$ coordinates in the environment. Points are colored based on their assigned latent cluster ID. The top row (VAE-WM) shows scattered, noisy clusters, indicating that spatially distant frames are incorrectly grouped together. The bottom row (GRWM) shows well-defined, spatially coherent clusters, demonstrating that our learned latent space is structurally aligned with the environment's true state manifold.

made progress toward this goal, but the challenge remains open. The path forward lies in developing methods that can not only learn a geometrically sound manifold, as we have done, but can also automatically disentangle the true factors of variation. A model equipped with such a perfect representation would not just be cloning an environment; it would be understanding its fundamental laws.

In conclusion, our work repositions the problem of high-fidelity world modeling as one of representation learning first, and dynamics modeling second. By shifting the focus from the complexity of the transition function to the geometry of the state space, we have taken a significant step toward building robust, long-horizon predictive models.

## ETHICS STATEMENT

This work does not involve human subjects, sensitive data, or potentially harmful applications. Our methods are intended for scientific research purposes only.

## REPRODUCIBILITY STATEMENT

All experiments presented in this paper are reproducible. The model architectures, training procedures, and hyperparameters are described in detail in the main paper and appendix. We will release our code and datasets upon acceptance.

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

# Appendix

## A  LLM Usage

LLMs were used only to polish language and improve writing fluency; all research content is solely by the authors.

## B  Ablation Studies

We conduct ablation studies to validate the contribution of our core components and design choices. Specifically, we examine four aspects: (1) the necessity of the two core loss terms, (2) the role of the projection head, (3) the impact of latent dimension, and (4) the effect of critical design choices on model performance.

### B.1  Importance of Core Regularization Terms

Both slowness and uniformity losses are essential and complementary. We evaluate four model variants: a vanilla VAE, the full model, and two partial variants without $\mathcal{L}_{\text{uniform}}$ or $\mathcal{L}_{\text{slow}}$. For rollout evaluation, both partial variants diverged and produced NaN values, so we only report their loss statistics from autoencoder training. For completeness, we report the values of all loss terms during autoencoder training, even when they are not directly optimized. As shown in Table 2, removing either regularization leads to a substantial drop in rollout performance.

Table 2: Ablation study on the effect of regularization terms. We report the reconstruction loss $\mathcal{L}_{\text{recon}}$, the monitored slowness loss $\mathcal{L}_{\text{slow}}$, and the monitored uniformity loss $\mathcal{L}_{\text{uniform}}$.

| Model | $\mathcal{L}_{\text{recon}}$ | $\mathcal{L}_{\text{slow}}$ | $\mathcal{L}_{\text{uniform}}$ |
|---|---|---|---|
| VAE-WM | 0.00042 | 0.88 | -3.04 |
| GRWM | 0.00067 | 0.11 | -3.13 |
| GRWM w/o $\mathcal{L}_{\text{uniform}}$ | 0.00052 | 0.00015 | 0 |
| GRWM w/o $\mathcal{L}_{\text{slow}}$ | 0.00100 | 0.46 | -2.47 |

When optimizing for slowness alone (w/o $\mathcal{L}_{\text{uniform}}$), we observe a classic case of representation collapse. The model aggressively minimizes the slowness loss by mapping all representations to a tiny region of the latent space, evidenced by a very high uniformity loss and low slowness loss. This confirms that $\mathcal{L}_{\text{uniform}}$ is indispensable for preventing trivial solutions.

When optimizing for uniformity alone (w/o $\mathcal{L}_{\text{slow}}$), we interestingly observe that the slowness metric naturally decreases. We attribute this effect to pushing different trajectories apart, which implicitly encourages representations from the same trajectory to cluster. However, explicitly including $\mathcal{L}_{\text{slow}}$ accelerates and reinforces this trend.

### B.2  Role of the Projection Head

Table 3: Effect of the projection head. The projection head reduces reconstruction loss while maintaining latent probing performance.

| Model | $\mathcal{L}_{\text{recon}}$ | Probing MSE |
|---|---|---|
| VAE-WM | 0.00039 | 0.136 |
| GRWM (w/ proj) | 0.00061 | 0.058 |
| GRWM (w/o proj) | 0.00291 | 0.054 |

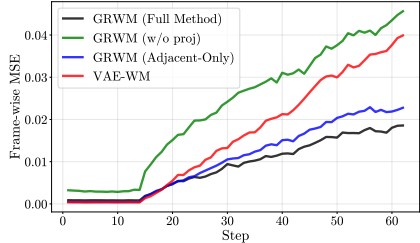

Figure 7: Frame-wise prediction MSE across ablation variants.

The projection head disentangles representation structuring from pixel-level reconstruction. While the model without a projection head can still learn a reasonably well-aligned latent space (as measured by latent probing MSE), its reconstruction loss is higher, as shown in table 3. By introducing

the projection head, we allow the regularization losses to act in a separate subspace, freeing the primary latent space $z$ to focus on accurate reconstruction. This decoupling ultimately leads to better overall predictive performance and lower frame-wise MSE, as shown in figure 7.

### B.3 ANALYSIS ON LATENT DIMENSION

We conducted an ablation study on the dimensionality of the latent space, testing dimensions of 16, 32, 64, and 128. The results is presented in Figure 8.

The benefits of our regularization are independent of the latent space size. GRWM consistently outperforms the vanilla VAE-WM across all tested dimensions. For every capacity, the rollout error of GRWM (solid lines) is significantly lower than that of the corresponding baseline (dashed lines).

More importantly, our method demonstrates remarkable robustness to this hyperparameter. The performance curves for our model with latent dimensions 16, 32, 64, and 128 are nearly indistinguishable, indicating that our regularization technique successfully structures the latent space and learns a compact representation of the true state manifold, regardless of the available capacity. In contrast, the baseline's performance is highly sensitive to the latent dimension. For the vanilla VAE-WM, a larger latent space appears to be detrimental, leading to faster error accumulation. Without proper regularization, a higher capacity latent space may overfit to irrelevant visual details or capture noise, which harms long-term prediction. Our method effectively mitigates this issue, ensuring stable and predictable performance, which is a highly desirable property for practical applications.

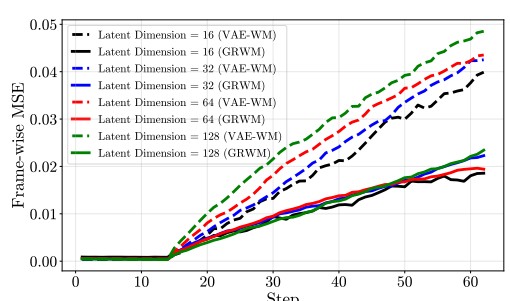

Figure 8: Ablation study on the impact of latent dimension. GRWM (solid lines) consistently and significantly outperforms the vanilla VAE baseline (dashed lines) across all tested latent dimensions (16, 32, 64, and 128). Notably, our method's performance is remarkably robust to the choice of latent dimension, while the baseline's performance is highly sensitive.

### B.4 DESIGN OF THE SLOWNESS LOSS

All-pairs temporal consistency is crucial for preventing degenerate solutions. A critical design choice in our $\mathcal{L}_{slow}$ formulation is to pull all pairs of frames within a trajectory's context window closer, rather than only adjacent pairs. We compare our "All-Pairs" approach with an "Adjacent-Only" baseline. The results show that the "All-Pairs" strategy is superior. We attribute this to the causal nature of our encoder. Since the encoder's output for frame $t$ is already conditioned on frames $t-k, \ldots, t-1$, simply minimizing the distance between $z_t$ and $z_{t-1}$ presents a "lazy" optimization problem due to their overlapping inputs. In contrast, our "All-Pairs" strategy enforces smoothness across distant frames with non-overlapping inputs (e.g., $z_t$ and $z_{t-k}$), leading to globally coherent latent trajectories and stronger long-horizon prediction.

## C ADDITIONAL ROLLOUT VISUALIZATIONS

As shown in Figure 9 our method significantly outperforms the VAE baseline: while the VAE predictions are already inaccurate at 100 steps, our model maintains high fidelity at this horizon. In some case, our method can occasionally produce accurate predictions even at 400 steps, demonstrating the improved consistency of the latent-space trajectories with the true environment. Importantly, these results are not cherry-picked; the figure shows samples from randomly selected starting points, illustrating the typical performance of both methods over long-horizon rollouts.

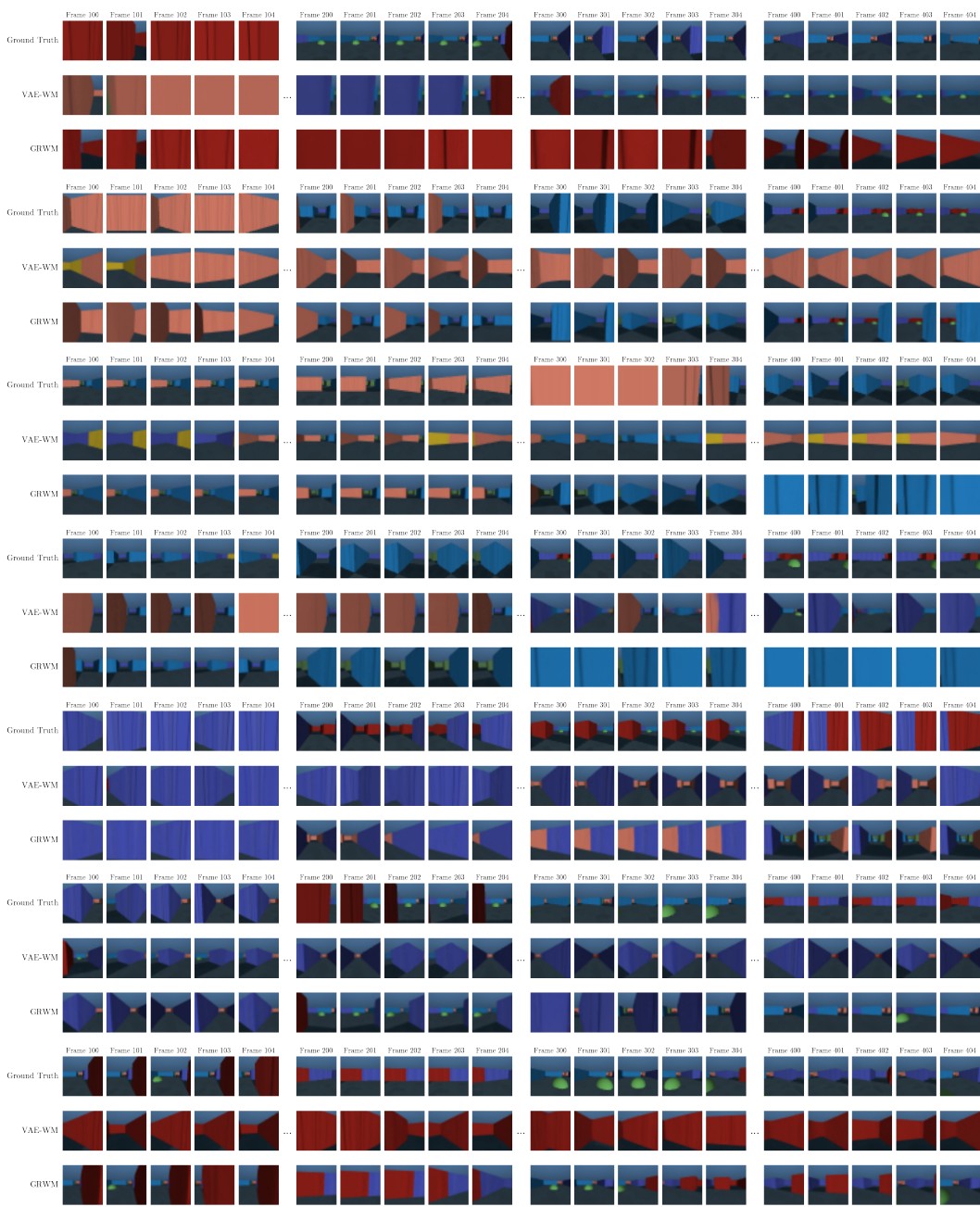

Figure 9: Visualization of generated frames at multiple time points from a single starting state. We show frames near steps 100, 200, 300, 400, sampled randomly — no cherry-picking. Our method significantly outperforms the VAE baseline: while the VAE predictions are already inaccurate at 100 steps, our model maintains high fidelity at this horizon. In some case, our method can occasionally produce accurate predictions even at 400 steps, demonstrating the improved consistency of the latent-space trajectories with the true environment.

# D    TRAINING DETAILS

We summarize the training configurations for both the AutoEncoder and the dynamics models. Unless otherwise specified, models are trained with Adam optimizer and warmup-based learning rate schedules. The complete set of hyperparameters, including environment-specific settings, is pro-

vided in Table 4. For reproducibility, we will release the code and all configuration files upon paper acceptance.

Table 4: Training hyperparameters for AutoEncoder and Dynamics models.

| AutoEncoder Training | Setting |
| --- | --- |
| Epochs | 50 |
| Optimizer | Adam (lr $5 \times 10^{-4}$) |
| Scheduler | Warmup-linear (1000 warmup, 10,000 total, min ratio 0.1) |
| Architecture | Layers [2, 1, 2, 2, 1, 1, 2]; Encoder channels [256, 32]; Channels [256, 512, 512] |
| Maze 9×9 | $\lambda_{\text{uniform}} = \lambda_{\text{slow}} = 0.1$, $\beta = 1 \times 10^{-6}$ |
| Maze 3×3 | $\lambda_{\text{uniform}} = \lambda_{\text{slow}} = 0.1$, $\beta = 1 \times 10^{-6}$ |
| Minecraft | $\lambda_{\text{uniform}} = \lambda_{\text{slow}} = 0.01$, $\beta = 1 \times 10^{-6}$ |

| Dynamics Models Training | Setting |
| --- | --- |
| Optimizer | Adam ($\beta_1 = 0.9$, $\beta_2 = 0.99$), weight decay $1 \times 10^{-4}$ |
| Scheduler | Warmup-decay (1000 warmup, 100,000 total steps) |
| Diffusion | 1000 steps, cosine $\beta$-schedule (shift=10.0), noise clip 20.0 |
| Objective | `pred_v`, fused-min-SNR loss weighting (clip=20.0, decay=0.9) |
| Sampling | DDIM, 5 steps, $\eta = 0.0$ |
| VDM | Classifier-free guidance, weight=5 |

# E  DATASET DETAILS

We evaluate our models on two environments: a memory Maze environment and a Minecraft environment.

**Maze.**  For the Maze environment, we fix the random seed to generate a consistent set of maps. We use Memory-Maze Environment (Pasukonis et al., 2022). The rendered images are obtained using the MuJoCo engine. The agent has a discrete action space consisting of {move forward, turn left, turn right}. Trajectories are collected with a noisy A* algorithm to ensure sufficient coverage of the maze.

**Minecraft.**  For the Minecraft environment, we adopt the map from Gornet & Thomson (2024). To restrict exploration, we enclose the area with wooden fences. Data collection is performed using the Malmo framework, with the same action space as in the Maze environment. The noisy A* policy is again used to generate diverse trajectories.

**Statistics.**  Each trajectory contains up to 1000 frames, though most consist of several hundred frames. Each dataset contains 5000 trajectories in total.

**Trajectory Visualization.**  To provide an intuitive understanding of the datasets, we visualize several representative trajectories. Figure 10 shows examples from three settings: M3x3-DET, M9x9-DET, and MC-DET.

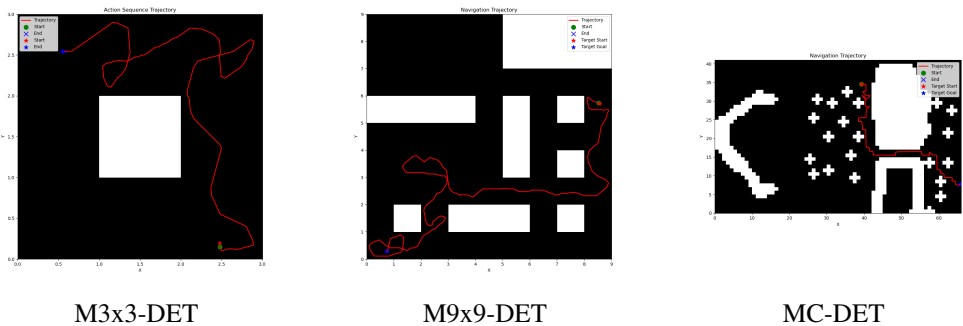

| M3x3-DET | M9x9-DET | MC-DET |

Figure 10: Representative trajectories from the three datasets. Each plot shows a sample trajectory overlaid on the environment layout.

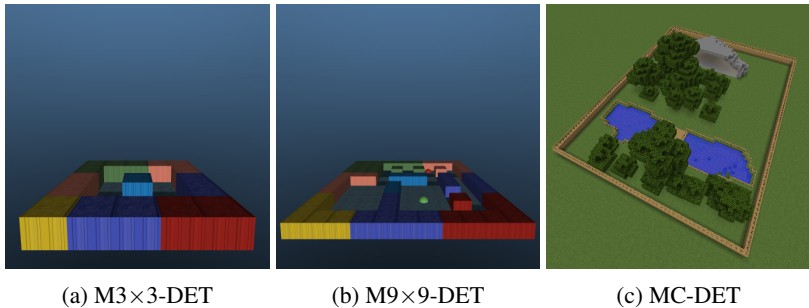

(a) M3×3-DET      (b) M9×9-DET      (c) MC-DET

Figure 11: High-angle perspective views of the three evaluation environments. These renderings provide an intuitive, three-dimensional understanding of the maze layouts that complements the 2D top-down maps in the main text.

