# OpenReview forum: "Clone Deterministic 3D Worlds with Geometrically Regularized World Models"
_ICLR.cc/2026/Conference — ICLR 2026 Conference Withdrawn Submission_

### Official Review · Reviewer_CNko · 2025-10-28

**Soundness:** 2
**Presentation:** 2
**Contribution:** 2
**Rating:** 2
**Confidence:** 5

**Summary:**

This paper addresses the challenge of poor long-horizon prediction fidelity in existing world models.

This paper focuses on high-fidelity cloning of deterministic 3D environments (e.g., fixed 3D mazes).

This paper proposes Geometrically-Regularized World Models (GRWM) to learn latent spaces aligned with the true state manifold of the environment.

GRWM consists of a temporal contextual architecture (causal encoder + instantaneous decoder) that integrates temporal context to resolve perceptual aliasing, and two geometric regularization losses.

The paper validates GRWM on three self-curated deterministic datasets showing it outperforms vanilla VAE-based baselines in long-horizon prediction.

**Strengths:**

**The research direction is important and valuable**: Existing world models often focus on random generation of open worlds, but neglect the need for high-fidelity modeling of deterministic scenarios (such as fixed-map mazes and static space robot navigation). This direction fills this research gap and is closely aligned with practical scenarios such as robot planning and game AI, providing a clear problem-oriented approach.

**This paper exhibits good writing**—with precise dataset details and fluent language—strong motivation (highlighting existing world models’ poor long-horizon prediction, the oversight of deterministic environments), and clear logic, forming a coherent and easy-to-follow research paper.

**Weaknesses:**

**The paper's statement that "Achieving precise world simulation with world models requires overcoming two tightly intertwined challenges" lacks persuasiveness, stemming from a lack of supporting evidence and insufficient citations.** From a logical perspective, the authors summarize the core challenges of "precise world simulation" as two key points, but neither empirically supports the centrality of these two challenges nor cite authoritative research or review literature in the field to establish consensus on their assertion. This unfounded assertion renders the key issues raised lacking credibility within an academic context. Furthermore, the lack of references in the entire Introduction further weakens the rigor of the argument. Extensive research in the world model field has addressed topics such as "degraded prediction accuracy due to defects in representation learning" and "the impact of coupling between dynamical modeling and representation quality on generalization." The authors' failure to engage with these classic or cutting-edge literature in dialogue with these topics undermines both the continuity of their research and the problem-solving and innovative nature of their work.

**The PRELIMINARY section of the paper is indeed thin in content and severely lacks citations, making it difficult to effectively connect with existing research in the field and undermining the innovation of the work.**
First, in the "Next-State Prediction Problem" section, the authors only define the next-state prediction objective and error calculation method for the world model, but fail to cite classic or cutting-edge research in this field. This lack of citations not only hinders the reader's understanding of the research context of the problem, but also hinders the author's subsequent proposal of "learning a representation space for linearized deconstruction dynamics" from being compared with existing methods, making its necessity difficult to demonstrate.
Secondly, the "Latent Generative Models" and "Contrastive Learning" sections are equally brief and lack supporting literature. Regarding latent generative models, the authors fail to cite the foundational work of other core models, nor do they introduce relevant research in the field that uses these models to improve the quality of world model representations. Regarding contrastive learning, the authors fail to explain the current application of existing contrastive learning methods in learning world model representations. This vague treatment of "previous work" makes it difficult for readers to clearly distinguish the differences between the GRWM proposed by the author and existing latent generation and contrastive learning methods, and thus makes it difficult for readers to understand the uniqueness and innovation of their contributions, which violates the core function of the PRELIMINARY section of "sorting out the foundation of the field and clarifying the research positioning."

I appreciate the authors' motivational experiments. **However, I'm puzzled by the fact that the contribution and novelty of this paper need to be clearly articulated.** Incorporating temporal context is a simple operation that has been extensively studied in other papers and fields. Adding a regularization term to the loss to constrain the latent space is also not a novel approach. I don't think this pipeline is innovative at all.

**Questions:**

Are there no other sota methods in this field? Why do the authors only make a detailed comparison with VAE-WM throughout the whole paper, but lack other methods?

All ablation studies are placed in the appendix. Why are they not placed in the main text? Do you think these ablation studies are not important? I think B.1 and B.2 are very important ablation studies and should be placed in the main text.

The total number of references is only about 40, which makes it difficult for me to believe that the authors have read extensively all the previous work in the field and made reasonable improvements on the issues they are missing.

---

### Official Review · Reviewer_ksfm · 2025-10-31

**Soundness:** 2
**Presentation:** 3
**Contribution:** 1
**Rating:** 2
**Confidence:** 4

**Summary:**

This paper proposes Geometrically-Regularized World Models (GRWM), consisting of a latent forward model and a sequence encoder trained by contrastive learning through time, aiming at improving long-horizon world model fidelity in deterministic 3D environments. The authors argue that the main performance bottleneck lies in the quality of the learned latent representations rather than the dynamics model. GRWM uses the well-known principle of contrastive learning through time which the authors call "temporal geometric regularization" based on their "temporal slowness" and "latent uniformity" losses. This leads to latent embeddings that are more likely to be similar for environment states on the same trajectories and less likely to be similar if they are not on the same trajectory. Experiments on deterministic maze and Minecraft-like datasets show consistent improvements over a simple vanilla VAE-based world model chosen by the authors.

**Strengths:**

- The research question is well-motivated: how representation quality affects long-horizon prediction stability.

- The use of deterministic environments is a good design choice to isolate representation from stochasticity.

- The empirical setup is consistent and the upper-bound "oracle" experiment is a useful reference point.

- The paper is generally well-organized and readable.

**Weaknesses:**

While the paper presents a coherent and technically solid application of known self-supervised and contrastive representation principles to deterministic world models, its conceptual novelty is limited, several claims are overstated, and important methodological details are unclear. Despite consistent quantitative improvements over the chosen VAE baseline, the contribution appears incremental rather than groundbreaking.

Major:

- Limited novelty: GRWM combines a sequence-based VAE encoder with temporal contrastive regularization (i.e. contrastive learning through time, which should be cited accordingly). Both components are well-established. Beyond packaging these elements for deterministic world modeling, there is nothing new.

- Potentially unfair baseline: It is unclear whether the "standard VAE" baseline also receives sequential inputs. If not, the comparison is confounded by the presence of temporal context, which alone can inject global information. A fair evaluation should compare against state-of-the-art sequence-based world models and VAEs trained on sequences with identical context windows. The Ablation studies do not include the case without L_slow and L_uniform, plus only the losses are compared, not the performance.


Minor:

- Overstated claims: Phrases like "necessitates linearization" and "aligned" vs. "misaligned latent space" are not sufficiently supported.

- Terminology: Terms such as "causal encoder/transformer" are used without clear definition or citation. If this is simply a causal mask with a sliding window, this should be clarified and cited. The term "geometric regularization" is misleading, as there is no explicit use of geometry or geometric concepts in the authors approach. Moreover, the writing occasionally uses inflated phrasing ("true topology of the environment", "fundamental laws", etc.), which detracts from technical clarity.

- Presentation: Section 3.3 introduces sequences of latent vectors with undefined shapes (B, L, "...") and includes unnecessary citations (e.g., for linear projection layers). Tighten notation and prune citations to essential ones.

- Position vs. observation: The oracle experiment’s interpretation should acknowledge that coordinates encode global state while images are local. This is an argument for using sequences as they can represent global state as a unique collection of local observations.

**Questions:**

- VAE Baseline: Does the vanilla VAE baseline receive identical temporal context and action conditioning? Specify the exact sequence length, masking, and action inputs per method. If not matched, please provide results under identical temporal context to isolate the effect of your regularization.

- Definition of "causal" modules: What precisely is meant by "causal encoder/transformer" here? If this is a standard causal mask with a sliding window, please say so and cite accordingly. If different, provide a concrete definition and implementation details.

- Relation to CLTT and novelty: Your slowness/uniformity objectives appear to be contrastive learning through time (CLTT). Please position your method relative to CLTT and closely related temporal contrastive/self-supervised approaches, and clarify what is substantively new in your losses or training protocol.

- Comparison: Why are state-of-the-art sequence-based world models or sequence-VAEs with identical context windows not included as baselines?

---

### Official Review · Reviewer_FhwT · 2025-10-31

**Soundness:** 3
**Presentation:** 3
**Contribution:** 2
**Rating:** 6
**Confidence:** 3

**Summary:**

This paper argues that the primary bottleneck for world models in faithfully cloning deterministic 3D environments is the quality of the latent representation, not the dynamics model. The authors introduce Geometrically-Regularized World Models (GRWM), a plug-and-play VAE modification that learns a latent space structurally aligned with the environment's true physical topology. This is achieved using two unsupervised losses: a Temporal Slowness Loss to cluster trajectory points and a Latent Uniformity Loss to separate them. Experiments show GRWM dramatically reduces long-horizon prediction error across multiple dynamics models by preventing the catastrophic "teleportation" and "mode collapse" artifacts seen in standard VAEs.

**Strengths:**

1. The "representation matters" thesis is clearly articulated and convincingly demonstrated via an "oracle" experiment (Fig. 1), which shows a dynamics model works perfectly on ground-truth states but fails on VAE latents .

2. GRWM is a lightweight, plug-and-play solution (two losses and a projection head) that makes it practical and broadly applicable to existing VAE-based world models .

3. The method significantly and consistently reduces long-horizon prediction error (Fig. 3) and, unlike baselines, avoids "teleportation" in ultra-long-horizon (10,000-step) rollouts (Fig. 5) .

**Weaknesses:**

1. The method is explicitly designed for and tested on static, deterministic 3D worlds. It's unclear how the core slowness principle would apply to stochastic environments.

2. The core components ($\mathcal{L}_{slow}$ and $\mathcal{L}_{uniform}$) are established principles from temporal slowness and contrastive learning, respectively10101010. The novelty is in their application to world model VAEs.

3. The 3D Maze and Minecraft environments are topologically simple. The method's robustness to visually complex textures or severe aliasing (beyond simple corridors) is not fully tested.

**Questions:**

1. How do you expect $\mathcal{L}_{slow}$ to perform in stochastic environments, where it might incorrectly penalize valid, sharp state changes?

2. Your "All-Pairs" $\mathcal{L}_{slow}$ is better than "Adjacent-Only" . Did you test other weightings, like an exponential decay, rather than a uniform average over the context window?

3. How sensitive is performance to the temporal context window size ($k$)?

4. How does an unsupervised GRWM compare to a supervised VAE that is given the ground-truth (x, y, $\theta$) states as a regression target during training? This would quantify how much of the "oracle gap" you closed without supervision.

---

### Official Review · Reviewer_foy8 · 2025-10-31

**Soundness:** 1
**Presentation:** 2
**Contribution:** 1
**Rating:** 0
**Confidence:** 5

**Summary:**

This paper considers that the world model bottleneck in long-term error accumulation lies in the representation quality, rather than the dynamical model itself. To address this, they propose a Geometrically Regularized World Model (GRWM), which introduces temporal context through a causal encoding structure and designs two self-supervised losses (temporal slowness and latent uniformity) to enhance the geometric structure of the latent space, making it more closely resemble the real-world manifold. This method can be directly combined with existing VAE representation modules and integrated with three diffusion-based dynamical models. Experiments validate the effectiveness of the method in three closed, deterministic environments (two maze scenarios and a Minecraft scene): GRWM significantly reduces error accumulation in long-term prediction and generates a latent space with a clearer topology and more consistent with the real-world state.

**Strengths:**

1. A simple and lightweight latent space regularization method is proposed, which can be directly integrated with existing world model architectures.

2. Experimental results show a reduction in relative medium-term prediction errors in deterministic environments.

3. Qualitative visualizations help to understand latent space failure modes.

4. The paper provides insights and evidence on the importance of representation learning in world models.

**Weaknesses:**

SOUNDNESS

1. MSE is the only metric used, but it cannot completely show the performance or prediction quality. For that, one needs to use other metrics: SSIM, PSNR, and even object-level errors. MSE in pixel space is not a reliable measure for dynamics fidelity or physical consistency. Additional downstream evaluation is required.

2. For the loss function, the pixel-level MSE is not equivalent to physical consistency.  For example, the \beta: the paper does not discuss the interaction effects that influence the geometric quality of latent objects. For the \lambda_slow: It can easily cause collapse, and the physical state may not change gradually. For \lambda_uniform: it may incorrectly exclude similar samples in the same state. It is unclear whether the uniformity objective contradicts the core goal of aligning with the true state manifold, since multiple occurrences of the same physical state in different trajectories are explicitly pushed apart.

3. The chosen baselines are not the SOTA: they do not include the most relevant or competitive representation learning approaches for sequential data, making the relative gains hard to interpret.

4. No statistical confidence/variation is reported, making it unclear whether improvements are stable across random seeds and trajectories. Cross-validation is also lacking, making it possible that the hyperparameters are overfit.

5. The probing task leverages ground-truth state labels, which contradicts the unsupervised premise and only measures linear predictability rather than true manifold alignment.

6. There is a gap between theoretical claims and evidence. The paper claims that "geometric alignment between the latent space and the real-state manifold" is the reason for the performance improvement, but it lacks quantifiable topological/geometric consistency metrics and analyses (such as trustworthiness/continuity, distance preservation/proximity preservation rate, manifold dimension estimation, geodesic distance preservation, and error comparison of linear regressibility to the real state). Current evidence is more about intuitive visualization and downstream error reduction, which is insufficient to support the strong claim of "aligned manifold".

7. The mutual interference of Loss was not explained or analyzed.

8."Causal encoding + single-frame reconstruction" may still lead to a "lazy" solution (which is what the paper intends to overcome).

9. Although 3.2 claims to prevent the model from only looking at the last frame, it lacks ablation proof that zt does utilize history.

10. The boundaries of innovation are unclear. GRWM's two constraints (time-gradual variation + uniformity) are derived from existing work: the "dispersion/uniformity" ideas of SimCLR/BYOL/VICReg/Barlow Twins. This paper needs to clearly explain the differences between these methods in terms of objective function and theory, and make a strong baseline comparison (see "Required Experiments").

GENERALIZATION

11. The experimental cases are similar and simple: closed environments with low visual and dynamical diversity, which limits the generality of the claimed improvements. The task setting is too narrow, and its extrapolation is questionable. All evaluations are conducted in deterministic environments, and most appear to be small mazes/static maps. This setting avoids the most challenging aspects of reality (partial observability + randomness + sensor noise).

12. Potential negative sampling and aliasing risks: Uniformity loss treats any pair across trajectories as a negative sample. In deterministic environments, "the same state can appear on different trajectories," and this design pushes identical states apart, conflicting with the "geometric alignment" objective. Mechanisms to avoid this conflict need to be clearly defined (e.g., hard negative sampling based on state proximity, pseudo-positive samples at the same coordinates, etc.), and failure cases and mitigation strategies should be provided.

13. The assumption behind slowness (continuous motion) may be violated in environments with abrupt changes in direction or high-speed dynamics, casting doubt on the generalization beyond the deterministic maze setting.

14. The evaluation horizon is relatively short.

PRESENTATION

15. The ablation description is too short (section 5). The ablation study does not sufficiently isolate the contributions of temporal context and geometric regularization, nor does it probe failure modes.

16. The conclusion acknowledges that manifold alignment is not achieved, undermining the paper's core claim that representation learning resolves the long-horizon fidelity issue.

17. The clustering visualization lacks quantitative support; manifold preservation metrics should be included.

18. The "aligned/unaligned" visualization in Figure 1 requires repeatable measurements to avoid relying solely on visual impressions.

19. In 3.4, "Minimal overhead" should be given a specific number (% training time/increase in GPU memory).

20. The terms "clone/overfit" are used together: Please clarify the boundaries and evaluation methods between "high-fidelity cloning" and "overfitting".

21. If the goal is "controllable simulation", the controllability based on motion perturbation should be reported (divergent predictability of different actions from the same starting point).

22. Figure 1 does not clearly show the difference between the latent spaces of Misaligned Latent and Aligned Latent.

**Questions:**

QUESTIONS TO ANSWER

1. What are the formulation differences between this combination and SFA, TCC, and BYOL/VICReg/Barlow Twins? Why is your combination better suited to the world model? Are there any rigorous controlled experiments and statistical significance tests?

2. Uniformity is achieved by using all negative values across trajectories: when the same physical state appears on different trajectories, it may be incorrectly rejected. Has state equivalence detection/nearest neighbor filtering/temperature scheduling been employed?

3. How do you quantify your "alignment topology"? Do you report trustworthiness/continuity, neighborhood preservation rate of Isomap/UMAP, error in linear regression to true coordinates, local Lipschitz/curvature, etc.?

4. In 3.2 Causal Windows for the k: How to control the complexity O(k^2) (all pairs within pairs)? Is it performance-sensitive? Is there an optimal window length trade-off with out-of-domain generalization?

5. Hyperparameters: β,λslow,λuniform. How robust is the method to changes in hyperparameters? Is there a unified set of values across datasets/backbones?

6. Is the dynamical model action-conditioned? This indicates whether actions are explicitly encoded (otherwise, separating z from a might compromise controllability).

7. Is the long-term evaluation a purely open-loop or intermittent teacher-mandated? Are the training and validation maps strictly separated to avoid the shortcut of "memorizing map textures"?

8. Computational overhead: How does the time/memory overhead of projection + all-pairs regularization affect throughput? How does it compare to strong contrast baselines (InfoNCE, VICReg)?

EXPERIMENTS TO ADD

1. Some strong baselines: Time-series versions of SFA/TCC/Time-Contrast/TCN/Temporal InfoNCE, BYOL/Barlow Twins/VICReg.  Accepted implementation of "VAE + contrastive regularization" rather than author-selected variants. Unified backbone and budget, reporting saliency.

2. Topological consistency metrics (direct evidence for core claims).  Trustworthiness/continuity, neighborhood preservation rate, global/local distance fidelity, manifold dimension/curvature.  Error of linear/kernel regression from z to the true state (coordinates, orientation)

3. Robustness and Generalization: more complicated environments need to be added.

4. Ablation and Diagnosis: Remove \lambda_slow, remove \llambad_uniform. Use only reconstruction/KL, with different windows k. Scramble history/remove last frame/only look at last frame, to check if we truly avoid "laziness".  Negative sampling strategy (cross-trajectory vs. same-trajectory is difficult to negative), temperature/scale sensitivity.

5. More reliable and diverse performance metrics (see weaknesses).

---

### Note · Authors · 2025-11-14

I have read and agree with the venue's withdrawal policy on behalf of myself and my co-authors.